# Bactericidal and Biocompatible Properties of Plasma Chemical Oxidized Titanium (TiOB^®^) with Antimicrobial Surface Functionalization

**DOI:** 10.3390/ma12060866

**Published:** 2019-03-15

**Authors:** Stefan Kranz, André Guellmar, Andrea Voelpel, Tobias Lesser, Silke Tonndorf-Martini, Juergen Schmidt, Christian Schrader, Mathilde Faucon, Ulrich Finger, Wolfgang Pfister, Michael Diefenbeck, Bernd Sigusch

**Affiliations:** 1Department of Conservative Dentistry and Periodontology, University Hospital Jena, An der Alten Post 4, 07743 Jena, Germany; Andre.Guellmar@med.uni-jena.de (A.G.); Andrea.Voelpel@med.uni-jena.de (A.V.); Tobias.Lesser@med.uni-jena.de (T.L.); Silke.Tonndorf-Martini@med.uni-jena.de (S.T.-M.); bernd.w.sigusch@med.uni-jena.de (B.S.); 2Innovent Technology Development, Prüssingstraße 27 B, 07745 Jena, Germany; js@innovent-jena.de (J.S.); cs@innovent-jena.de (C.S.); 3Königsee Implantate GmbH, OT-Aschau—Am Sand 4, 07426 Allendorf, Germany; Mathilde.Faucon@koenigsee-implantate.de (M.F.); Ulrich.Finger@koenigsee-implantate.de (U.F.); 4Department of Microbiology, University Hospital Jena, Erlanger Allee 101, 07747 Jena, Germany; Wolfgang.Pfister@med.uni-jena.de; 5Department of Trauma-, Hand- and Reconstructive Surgery, University Hospital Jena, Erlanger Allee 101, 07747 Jena, Germany; Michael.Diefenbeck@bonesupport.com

**Keywords:** titanium implants, dental implants, antibacterial coating, gentamicin, silver, zinc, cytotoxicity, MC3T3-E1, Staphylococcus aureus, plasma chemical oxidation

## Abstract

Coating of plasma chemical oxidized titanium (TiOB^®^) with gentamicin-tannic acid (TiOB^®^ gta) has proven to be efficient in preventing bacterial colonization of implants. However, in times of increasing antibiotic resistance, the development of alternative antimicrobial functionalization strategies is of major interest. Therefore, the aim of the present study is to evaluate the antibacterial and biocompatible properties of TiOB^®^ functionalized with silver nanoparticles (TiOB^®^ SiOx Ag) and ionic zinc (TiOB^®^ Zn). Antibacterial efficiency was determined by agar diffusion and proliferation test on Staphylocuccus aureus. Cytocompatibility was analyzed by direct cultivation of MC3T3-E1 cells on top of the functionalized surfaces for 2 and 4 d. All functionalized surfaces showed significant bactericidal effects expressed by extended lag phases (TiOB^®^ gta for 5 h, TiOB^®^ SiOx Ag for 8 h, TiOB^®^ Zn for 10 h). While TiOB^®^ gta (positive control) and TiOB^®^ Zn remained bactericidal for 48 h, TiOB^®^ SiOx Ag was active for only 4 h. After direct cultivation for 4 d, viable MC3T3-E1 cells were found on all surfaces tested with the highest biocompatibility recorded for TiOB^®^ SiOx Ag. The present study revealed that functionalization of TiOB^®^ with ionic zinc shows bactericidal properties that are comparable to those of a gentamicin-containing coating.

## 1. Introduction

Surface functionalization mainly aims at increasing the osseointegration of orthopedic and dental implants by supporting the adherence of endogenous cells [1,2,3]. Besides commonly used techniques such as coating of the implant surface with hydroxyapatite [4], growth factors [5], or bisphosphonates [6], our group recently showed that plasma chemical oxidation (PCO) also is of great potential [7,8]. In the electrochemical process of PCO, the naturally occurring oxidation layer on titanium is converted into a ceramic-like surface of high porosity with up to four micrometers in thickness (so called bioactive TiOB^®^ surface). As observed by Diefenbeck et al., titanium implants modified by PCO are especially well osseointegrated with high bone-to-implant contact rates [7].

On the one hand, functionalization is applied to improve healing and incorporation of implants, but on the other, it can also be used to render the surface antimicrobial self-active. Studies have proven that implants with antimicrobial surface activity are capable of preventing microbial adherence efficiently, without affecting the viability of endogenous cells like osteoblasts [1,2]. Antimicrobial surface functionalization mainly aims at reducing the risk of infection-associated implant failure [9]. However, in cases of implant-associated infections, Staphylococcus spp. are considered the predominant bacteria, with Staphylococcus aureus often being the main causative agent [10,11].

In this context, our group clearly showed that a gentamicin-containing coating on TiOB^®^ implants sufficiently prevents Staphylococcus aureus from surface colonization in an osteomyelitis study in rats [12].

In times of increasing drug resistance and antibiotic-related events, the development of alternative and novel antibacterial approaches that are not dependent on the action of traditional antibiotics is of major concern [13]. Therefore, the present in vitro study focuses on the preparation and the antibacterial as well as biocompatible testing of bioactive TiOB^®^ surfaces functionalized with silver nanoparticles and ionic zinc.

The antimicrobial action of silver has been known since historic times, and silver particles have already been applied in many biomedical applications such as wound dressings [14], catheter systems [15], bone cements [16], and implant coatings [17] due to their bactericidal properties.

Application of zinc is another very common strategy to prevent microbial colonization and biofilm formation [18] and has already been evaluated in association with bioactive glass and bioceramics [19,20]. Currently, Zn is intensively studied in biodegradable metal alloys with self-active antimicrobial characteristics as well [21].

The focus of the present study is to analyze the bactericidal and biocompatible properties of bioactive TiOB^®^ functionalized with silver nanoparticles and ionic zinc. Since functionalization of TiOB^®^ implants with coatings of gentamicin-tannic acid (TiOB^®^ gta) already proved its antimicrobial efficiency in vivo [12], it was used as control in the present study.

## 2. Materials and Methods

### 2.1. Sample Preparation

Samples were manufactured from surgical grade titanium TiAI6V4 alloy rods (DIN ISO 5832-3) by Königsee Implantate GmbH, Aschau, Germany. Two different sample geometries were prepared: Cylindrical pins (active area: 12 mm long, 1.5 mm in diameter, n = 48 per TiOB^®^ surface functionalization) were used for the microbiological tests, and disk-shaped samples (2 mm thick, 15 mm in diameter, n = 6 per TiOB^®^ surface modification) for the cytotoxicity tests (Figure 1a,b). All samples were uniformly shot-blasted with aluminum oxide abrasives and subsequently with RKSP 120 ceramic particles (Rösler Oberflächentechnik GmbH, Untermerzbach, Germany).

TiOB^®^ surfaces were established by plasma chemical oxidation (PCO) as described earlier [7,8]. In brief, PCO is an anodic oxidation-based modification process of the naturally occurring oxidation layer found on titanium causing the formation of a ceramic-like, macro-porous, bioactive surface called TiOB^®^. The quality of the produced TiOB^®^ surface is dependent upon the applied electrolyte, the provided anode material, and the current and voltage curves used. PCO was performed by INNOVENT e.V., Jena, Germany.

### 2.2. Antibacterial Functionalization of TiOB^®^

Antibacterial functionalization of TiOB^®^ with gentamicin-tannic acid (TiOB^®^ gta), silver nanoparticles (TiOB^®^ SiOx Ag), and ionic zinc (TiOB^®^ Zn) was performed by INNOVENT e.V., Jena, Germany.

For visualization, all functionalized TiOB^®^ surfaces were characterized by scanning electron microscopy (SEM). Therefore, samples were subjected to carbon vapor coating and afterwards imaged using a ZEISS Supra 55 VP (Carl Zeiss Microscopy GmbH, Jena, Germany) equipped with InLens-SE-detector (Carl Zeiss Microscopy GmbH, Jena, Germany) and Everhard-Thornley-SE-detector (Carl Zeiss Microscopy GmbH, Jena, Germany), both driven in HV-mode.

#### 2.2.1. TiOB^®^ Gta

Functionalization of bioactive TiOB^®^ with gentamicin-tannic acid was performed exactly as described by Diefenbeck et al. [12]. In brief, tannic acid of pharmaceutical purity (Ph. Eur.) was used and bound to gentamicin by neutralization of a gentamicin-2 H_2_SO_4_ solution with NaOH causing precipitation of gentamicin-tannin complexes. Subsequently, 500 mg of the precipitated gentamicin-tannin complex was dissolved in formic acid (Ph. Eur.) and applied onto TiOB^®^ samples by dip coating. Immediately afterwards, samples were dried by volatilization of the solvent, leading to the formation of a crystalline gentamicin-tannic acid layer on top of the TiOB^®^ surface with 5 mg of gentamicin in total.

Because TiOB^®^ gta already proved an efficient bacteriostatic effect in vivo, it served as a control in the present study.

#### 2.2.2. TiOB^®^ SiOx Ag

For functionalization of bioactive TiOB^®^ with silver nanoparticles, all samples were primarily embedded into silicon and afterwards subjected to atmospheric-pressure plasma-assisted chemical vapor deposition (APCVD), as described by Beier et al. [22].

In brief, silver nanoparticles were deposited on TiOB^®^ in the presence of vaporized hexamethyldisiloxane (HMDSO) (Sigma-Aldrich Chemie GmbH, Taufkirchen, Germany) at room temperature under atmospheric pressure (2–6 bar). As the plasma source, a commercially available BLASTER MEF system (TIGRES, Rellingen, Germany) was used. Via an atomizer, vaporized HMDSO was sprayed into cold low-pressure plasmas in the presence of water-free air, leading to the deposition of silicon oxide (SiOx) on the TiOB^®^ surface. Simultaneously, a 5% silver nitrate solution (purity 99.8%, Merck, Darmstadt, Germany) was additionally sprayed into the plasma, causing the incorporation of silver nanoparticles into the growing SiOx layer. In each run, a SiOx Ag layer of 25 nm in thickness was formed on the macroporous TiOB^®^ surface.

#### 2.2.3. TiOB^®^ Zn

TiOB^®^ Zn was formed by two consecutive PCO steps. First, primary TiOB^®^ coating was conducted in a calcium phosphate electrolyte up to an end-point voltage of 200 V. Subsequently, the samples were rinsed with deionized water and dried with compressed air.

In a second step, PCO was carried out in a electrolyte containing 0.14 mmol/L ammonium hydrogenphosphate, 13.4 mmol ammonia solution (25%), and 0.5 mmol/L zinc ions (Zn(CH_3_CO_2_)_2_) stabilized by chelating agents. During the second PCO treatment, zinc ions were incorporated into the TiOB^®^ surface at an end-point voltage of 350 V.

### 2.3. X-ray Photoelectron Spectroscopy (XPS)

The presence of zinc and silver in the respective sample surface was verified by surface-sensitive XPS (Theta Probe, Thermo VG Scientific, Paisley, UK). The detection depth of the XPS measurement was approximately 10 nm. A depth profile analysis was established by local sputtering of the surface with an ion beam and successive measurements. The spectrometer used was a Theta Probe (Thermo VG Scientific, Paisley, UK) with monochromatic Al Kα radiation (1486.6 eV). Excitation (100 W) was performed at a voltage of 15 kV and an emission current of 6.7 mA in a raster area of 400 μm in diameter at 10^−9^–10^−8^ mbar. The spectrometer was equipped with an ion gun, and sputtering was performed with argon gas at approximately 1.5 × 10^−7^ mbar. The ions produced were then accelerated to 3 keV in a drift section at an ion current of approximately 1 µA. The focal spot of the ion beam was 4 mm² and, with a sputter yield of 1.39, led to a Ti removal of approximately 0.1 nm/s For sputtering of non-conducting samples, an electron gun was additionally used to prevent electrostatic charging during sputtering and measurement. For this purpose, electrons were accelerated onto the samples by applying 6 eV and 15 µA. Before analysis of a spectrum, the energy axis was calibrated on the basis of the O1s peak (EB(O 1s) = 531.0 eV). The quantitative analysis was performed using the peak areas and considering a background correction according to Shirley.

### 2.4. Experimental Section

For the following in vitro tests, the samples were assigned to four experimental groups (Table 1). All samples were sterilized by autoclaving (35 min at 134–138 °C and 2.16 × 105 Pa) before use.

#### 2.4.1. Bactericidal Properties of Functionalized TiOB^®^

For all antibacterial experiments the gram-positive species Staphylococcus aureus subsp. Rosenbach (ATCC 49230) was used. The species was cultivated aerobically on TS agar (tryptose–soybean agar, Oxoid, Wesel, Germany) at 37 °C. Test batches were arranged by suspending 2–3 colonies in TS broth (tryptose–soybean broth, Oxoid, Germany) followed by cultivation for approximately 4 h at 37 °C. Bacteria were harvested in the exponential phase of growth.

##### 2.4.1.1. Agar Diffusion Test

In this subsection the release of antibacterial active components from the functionalized TiOB^®^ surfaces was observed by agar diffusion test. Therefore, functionalized TiOB^®^ samples of cylindrical shape were placed in microplates (Figure 1a) and incubated in 300 µL PBS at 37 °C. After 2, 4, 6, 12, 24, and 48 h of incubation, eluates from 8 samples per TiOB^®^ surface were collected and stored at −20 °C until use.

For testing, TS agar was supplemented with proliferating Staph. aureus and poured into sterile petri dishes. After solidification of the agar, 100 µL of the stored eluates were pipetted into punched holes of 9 mm in diameter. The correspondent inhibition zones were analyzed after incubation for 24 h at 37 °C.

##### 2.4.1.2. Proliferation Test

The antibacterial efficiency of the functionalized TiOB^®^ surfaces was assessed by a proliferation assay adopted from Bechert et al. [23]. In brief, the test is based upon the capability of adherent Staph. aureus to proliferate.

Therefore, 8 samples of each TiOB^®^ surface were placed in a 96-well microplate using a customized holding device (Figure 1a). Each sample was then incubated with 300 µL bacterial solution (OD_546_ 0.5—equivalent to approximately 10^6^–10^7^ bacterial cells/mL) for one hour under slight shaking at 37 °C. Subsequently, the samples were removed, and non-adherent bacterial cells were discharged by washing twice with 100 µL PBS. Afterwards the rinsed samples were placed into 300 µL/well freshly prepared nutrient-reduced medium (PBS, 1% sterile TS broth, 0.25% glucose, 0.2% (NH_4_)2SO_4_)) and incubated for another 18 h. Subsequently, all samples were removed and fresh TS broth at a ratio of 1:3 was applied. Optical density (OD540 nm) was measured at baseline and after incubation for 1, 2, 3, 4, 5, 6, 7, 8, and 24 h at 37 °C.

In a second part, all functionalized TiOB^®^ samples were primarily aged (pre-incubated) for 2, 4, 6, 12, 24, and 48 h in PBS. The proliferation test was then performed in the same way as described above.

#### 2.4.2. Biocompatible Properties of Functionalized TiOB^®^

In this part of the experiment, the viability of MC3T3-E1 cells (DSMZ ACC 210) was determined after direct cultivation on top of the functionalized TiOB^®^ surfaces for 2 and 4 d.

Cells were cultivated in alpha-MEM (minimal essential medium) supplemented with 10% FBS (fetal calf serum) and 1% PenStrep (penicillin-streptomycin) (all GIBCO; Karlsruhe, Germany) at 37 °C with 5% of CO_2_. Confluent cells at a density of 4.500 cells/cm^2^ were transferred to 12-well plates and cultivated on the top of disk-shaped TiOB^®^ samples in 2 mL α-MEM (37 °C, 5% CO_2_) for 2 d or 4 d.

Afterwards, the samples were washed twice with 2 mL PBS and each labeled with freshly prepared staining solution (12 µL fluorescein diacetate (FDA)/16 µL ethidium bromide in 2 mL PBS). Labeled cells were imaged using a fluorescence microscope (Nikon, Labophot, Japan, 10 × phase-contrast objective, λex = 455–495 nm). Viable cells appeared in green whereas non-viable cells showed red fluorescence. Nine fields of view per sample were examined with regard to live/dead cells, with three repetitions per TiOB^®^ surface.

### 2.5. Statistics

The data were analyzed using SPSS 19.0 for Windows (IBM, Armonk, NY, USA). Statistical significance in the proliferation test was determined by one-way ANOVA, addressed by Bonferoni correction to diminish accumulation of alpha errors. The level of significance was set to *p* < 0.05.

The results of the cytotoxicity test were graphically displayed by boxplots. Significant differences in the cell counts were checked by t-test.

## 3. Results

### 3.1. Surface Characterization by Scanning Electron Microscopy

SEM observation revealed different surface configurations (Figure 2a–d). In Figure 2a, the surface topography of non-functionalized bioactive TiOB^®^ (control) is shown. Bioactive TiOB^®^ without antibacterial functionalization is characterized by a specific macroporous surface and described elsewhere in detail [7,8]. In contrast, functionalization of TiOB^®^ with gentamicin-tannic acid (TiOB^®^ gta, Figure 2b) resulted in a rather smooth surface topography. In the case of TiOB^®^ gta, the entire macroporous TiOB^®^ structure is covered by a thick layer of gentamicin-tannic acid. A similar appearance is found for samples functionalized with ionic zinc (TiOB^®^ Zn, Figure 2d). TiOB^®^ Zn is characterized by smooth surface areas interrupted by some large pores. In Figure 2c, the surface of TiOB^®^ functionalized with silver nanoparticles (TiOB^®^ SiOx Ag) is shown. Here, the sample surface is similar to non-functionalized bioactive TiOB^®^ and also of macroporous appearance.

### 3.2. XPS Analysis

Samples functionalized with gentamicin-tannic acid by dip coating showed a brownish crystalline cover layer with 5 mg gentamicin per sample.

The TiOB^®^ SiOx Ag samples revealed a layer of approximately 100 nm in thicknesses with a content of 5% silver incorporated into the silicon oxide matrix (Figure 3a).

In the case of the TiOB^®^ Zn samples, a matrix layer of 13 µm in thickness was produced with high concentrations of zinc in the top layers. In detail, the content of Zn varied from 20% in the top layer to 7% at a depth of 160 nm (Figure 3b).

### 3.3. Bactericidal Properties of Functionalized TiOB^®^

The release of gentamicin and active Ag and Zn from the respective functionalized TiOB^®^ surfaces was analyzed by means of an agar diffusion test.

It was shown that only eluates obtained from TiOB^®^ functionalized with gentamicin-tannic acid were able to inhibit the growth of Staph. aureus (Table 2). In case of TiOB^®^ gta, inhibition zones were observed during the entire study period (48 h) with the largest zones recorded at baseline.

In contrast, eluates collected from the surfaces functionalized with Ag NPs or ionic Zn completely failed to induce any inhibition zones in the agar diffusion test. TiOB^®^ without any antibacterial functionalization (TiOB^®^ control) also showed no signs of bacterial inhibition (Table 2).

In the second part of the examination, the capability of Staph. aureus to proliferate on the functionalized surfaces was observed (Figure 4). It was shown that all functionalized TiOB^®^ surfaces caused reduced rates of bacterial proliferation accompanied by extended lag phases (Figure 4). For TiOB^®^ gta, a lag phase of 5 h was detected, whereas TiOB^®^ SiOx Ag and TiOB^®^ Zn showed lag phases of 8 and 10 h. Further, the bacterial solutions obtained from Staph. aureus grown for 24 h in direct contact with the antibacterial surfaces were also of less optical density compared to those arranged from the TiOB^®^ controls (blue graphs in Figure 4).

Moreover, it was found that the antibacterial activity of all functionalized TiOB^®^ surfaces was dependent upon the time of pre-incubation in PBS (aging).

Among all surfaces examined, the strongest antibacterial effect was observed for samples that were not incubated in PBS prior to examination (presented by red graphs in Figure 4). In the cases of TiOB^®^ gta and TiOB^®^ Zn, significant antibacterial activity was still detected after pre-incubation in PBS for 24 h (Figure 4a,c).

However, in the case of the TiOB^®^ SiOx Ag surfaces, significant bacterial suppression was detected only for samples that were incubated in PBS for 2 and 4 h (Figure 4b).

### 3.4. Biocompatible Properties of Functionalized TiOB^®^

In Figure 5, the results of the LSM observation are shown. As expected, non-functionalized TiOB^®^ (TiOB^®^ control) was colonized by viable MC3T3-E1 cells to the highest extent (Figure 5 and Figure 6). Even after only 2 days of cultivation, the majority of cells exhibited a typical spread-like cell shape (Figure 5). In the case of the functionalized surfaces, best cytocompatibility was observed for TiOB^®^ SiOx Ag and TiOB^®^ Zn. In detail, after 2 days of cultivation, the largest number of viable cells, with an equal share in non-viable cells, was observed for TiOB^®^ SiOx Ag (Figure 6a). Similar to the TiOB^®^ control, viable cells grown on TiOB^®^ SiOx Ag showed a healthy spread-like shape (Figure 5). However, after 2 days of cultivation, the number of healthy cells grown on TiOB^®^ gta and TiOB^®^ Zn was rather low when compared to the TiOB^®^ control (Figure 5). On these surfaces, the majority of viable cells were of round or spheroidal shape (Figure 5).

After 4 days of cultivation, a significant increase in viable cells was observed for TiOB^®^ SiOx Ag and also TiOB^®^ Zn, with healthy cells grown on the entire surface (Figure 5 and Figure 6b). Unfortunately, no significant increase in viable cells was witnessed for TiOB^®^ gta (Figure 6b). Even after 4 days of cultivation, the majority of cells remained of round and spheroidal shape (Figure 5).

## 4. Discussion

In dental and orthopedic implant-based surgery, postoperative infections with pathogenic bacteria often cause serious complications [10]. Various studies have shown that implants with antibacterial surface activity are capable of preventing microbial colonization [24]. However, the major challenge is still to combine significant bacterial growth inhibition with sufficient biocompatibility [25].

The present study aims to determine the bactericidal and biocompatible properties of plasma chemical oxidized titanium functionalized with coatings of gentamicin-tannic acid, silver nanoparticles, and ionic zinc.

As shown by our group already, colonization of TiOB^®^ surfaces by *Staph. aureus* can sufficiently be prevented by a gentamicin-tannic acid coating. Early cell response towards TiOB^®^ gta has not yet been evaluated in detail.

However, the results of the present study revealed a reduced biocompatibility of TiOB^®^ gta. In contrast to uncoated TiOB^®^ (TiOB^®^ control), which was sufficiently colonized by MC3T3-E1, the majority of cells grown in direct contact with TiOB^®^ gta remained of round and spheroidal cell shape even after cultivation for 4 d.

Usually, gentamicin eluted from implant coatings is of good local and systemic biocompatibility [26]. This is also supported by findings of Popat et al. who observed no cytotoxic effect of gentamicin eluted from functionalized implant surfaces on MC3T3-E1 cells. Moreover, the authors realized that the characteristics of the surface seem to play a distinct role in early cell response [27]. The influence of cell attachment and proliferation behavior by topography features has already been studied by many other authors [28].

In the present study, SEM observations revealed a rather smooth surface of TiOB^®^ gta with complete coverage of the entire bioactive macro-porous TiOB^®^ structure, which might be one of the reasons for the reduced biocompatibility. Besides the unfavorable microstructure of the TiOB^®^ gta surface, viability might also be negatively influenced by the eluted tannic acid. In this context, Sahiner et al. clearly showed that an increase in tannic acid concentration is directly accompanied with a loss in viability of A549 cancerous cells and L929 fibroblasts [29].

However, as previously shown by our group, the applied gentamicin-tannic acid coating is dissolvable in water with partial exposure of the native TiOB^®^ surface after elution for 72 h [12].

We conclude that the rather smooth surface topography in combination with a concentration-dependent cytotoxic effect of the eluted tannic acid are responsible for the reduced biocompatibility observed for TiOB^®^ gta during the first four days of direct cultivation.

On the other hand, the applied gentamicin-tannic acid coating most efficiently suppressed Staph. aureus in the present study. Especially during the initial phase (<4 h), large inhibition zones were observed in the agar diffusion assay. Vester et al. also observed a rapid release of gentamicin from titanium implants coated with poly(D,L-lactide) [30].

Other groups have shown that an initial burst release of gentamicin by 40% within the first hour, 70% within the first 24 h, and 80% within the first 48 h is associated with good clinical, laboratory, and radiological outcomes [26]. More specific information regarding the active release of gentamicin from TiOB^®^ gta has been published by Diefenbeck et al. [12].

In general, biocompatibility of materials and drugs used in dentistry and surgery is different and highly dependent upon the type of substance or composition applied [31,32,33].

One major aim of the present study was to propose different antimicrobial functionalization strategies that are not dependent on the action of traditional antibiotics combined with surface characteristics better tolerated by endogenous cells. As shown by SEM observation, plasma-assisted chemical vapor deposition (PACVD) of silver nanoparticles (Ag NPs) resulted in surface topography features similar to those of the TiOB^®^ control. In comparison to TiOB^®^ gta or TiOB^®^ Zn, strong colonization by viable MC3T3-E1 cells was already observed after 2 days of direct cultivation.

So far, implants functionalized by silver nanoparticles have often shown reduced biocompatible properties which are caused by either the release of silver ions in high concentrations or by the small size of the particles. A moderate cytotoxic effect was confirmed by Kheur and coworkers, who reported a biocompatible response of only 69% in viable cells when primary human gingival fibroblasts were incubated in the presence of titanium sputtered with Ag for 5 min [34]. Further, Smeets et al. proved that Ag/SiOxCy-coated titanium implants were significantly less osseointegrated compared to grit-blasted or acid-etched implants [35].

In the present study, a significant increase in viable and spread-out cells was observed after 4 d of direct cultivation on TiOB^®^ SiOx Ag. This was directly accompanied by a decrease in bactericidal activity. As shown in the proliferation assay, significant bacterial inhibition of TiOB^®^ SiOx Ag only lasted for a maximum of 4 h. Unlike TiOB^®^ gta, no inhibition zones were observed in the agar diffusion test either. These results are in line with findings of Lisher et al., who also observed strong antibacterial effects of minimal Ag-containing plasma polymer coatings (Ag/amino-hydrocarbon) during the first day of immersion in deionized water [36].

When Ag NPs are oxidized, ionic Ag will be released with strong bactericidal activity [37]. Although the mechanisms of action are not fully understood, Ag has been applied as an antimicrobial agent since historical times. One reason for the antibacterial effect is found in the ability of silver ions to interact with essential thiol (sulfhydryl) groups causing a loss in bacterial enzyme function [38,39]. Further, Jaiswal et al. [39] and Feng et al. [38] discussed an inhibitory effect of Ag ions on microbial DNA replication and ATP synthesis. In addition, silver ions and Ag NPs are also likely to interact with phospholipids, leading to disturbance of the bacterial cell wall integrity [40,41]. Further, Ag NPs might trigger the formation of highly reactive oxygen radicals which are known to cause oxidative damage to DNA and proteins [42].

A recent study reported that the minimal inhibitory concentration (MICs) of Ag NPs on Staph. aureus is 300 µg/mL and the minimum bactericidal concentration (MBC) is 600 µg/mL, whereas 30 mg/mL are required for complete bacterial killing [43]. In order to increase the antibacterial efficiency of TiOB^®^ SiOx Ag, the content in Ag NPs has to be increased which then will certainly be accompanied by an undesired decrease in biocompatibility.

Bioactive TiOB^®^ was also functionalized by ionic zinc applied in two consecutive PCO steps. Similar to TiOB^®^ gta, the macroporous surface of non-functionalized TiOB^®^ was entirely covered. SEM observation revealed an implant surface with smooth areas interrupted by some large pores colonized by viable MC3T3-E1 cells with spread-like shape.

Zn is currently intensively studied in biodegradable metal alloys and, overall, presents good biocompatible properties. Investigations by Zhu et al. have shown that adhesion and proliferation of human bone marrow mesenchymal stem cells on biomedical metals can be increased by the additional application of Zn. Obviously, Zn plays a distinct role in the activation of several different intracellular pathways associated with gene activation, regulation, cell growth, differentiation, extracellular matrix mineralization, and osteogenesis [44]. Further, Yu and co-workers recently studied the biological properties of Zn incorporated in micro/nano-textured titanium surfaces by high-current anodization and found a positive effect on the proliferation behavior and alkaline phosphatase activity of osteoblasts [45]. In addition, Shen et al. also confirmed that Zn incorporated into coatings positively affects the proliferation and differentiation behavior of osteoblasts, resulting in enhanced peri-implant bone formation [46]. To the contrary, Pagano et al. showed a decrease in the cell number when human gingival fibroblasts and human keratinocytes were incubated with extracts obtained from glass ionomer cements modified with 6 wt% zinc L-carnosine [47]. In our study, the Zn coating showed high bactericidal activity towards Staph. aureus, which lasted for up to 48 h. Comparable to TiOB^®^ SiOx Ag, no inhibition zones were discovered in the agar diffusion test. Although the antibacterial mechanisms are still not fully understood, a damage by direct or electrostatic interaction with the cell surface or by oxidation due to the formation of highly reactive oxygen species such as 1O_2_, HO·, or H_2_O_2_ have been discussed [48,49]. Similar to the gentamicin-tannic acid coating, the Zn-functionalization still showed an inhibitory effect, even after pre-incubation in PBS for 48 h.

## 5. Conclusions

All antibacterial self-active TiOB^®^ surfaces observed in the present study proved a significant antibacterial effect on Staph. aureus with different biocompatible properties. Since gentamicin is able to elute out of TiOB^®^ gta, inhibition of bacterial growth in the peri-implant region is possible, whereas the antibacterial effect of TiOB^®^ SiOx Ag and TiOB^®^ Zn is restricted to only the surface. The highest initial biocompatibility of all functionalized surfaces was observed for TiOB^®^ SiOx Ag, which also proved the best-structured surface topography. TiOB^®^ Zn showed an antibacterial efficiency comparable to that of TiOB^®^ gta, with good biocompatible aspects after four days of direct cultivation. Further examinations are needed to also observe the performance of these functionalized TiOB^®^ surfaces in vivo.

## Figures and Tables

**Figure 1 materials-12-00866-f001:**
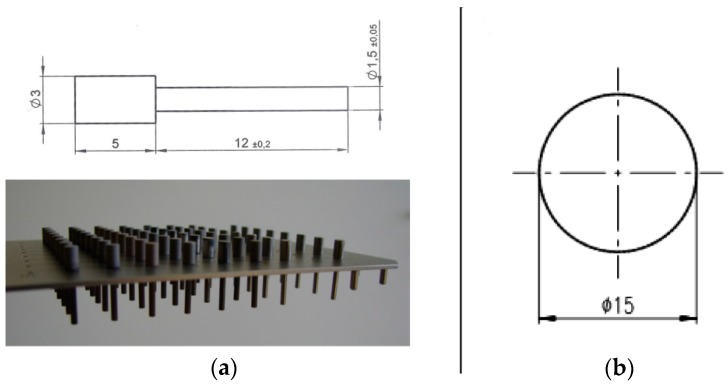
(**a**) Customized holding device and geometrical dimensions of the cylindrical samples used in the proliferation assay; (**b**) design of the disc shape samples used in the cytotoxicity tests.

**Figure 2 materials-12-00866-f002:**
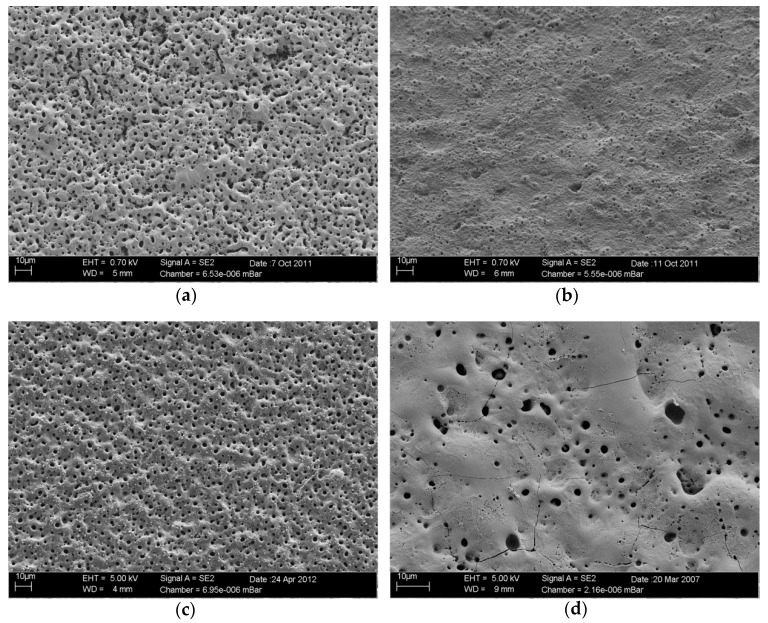
Surface characterization by scanning electron microscopy: (**a**) bioactive macroporous TiOB^®^ (TiOB^®^ control); (**b**) TiOB^®^ functionalized with gentamicin-tannic acid (TiOB^®^ gta); (**c**) TiOB^®^ functionalized with silver nanoparticles (TiOB^®^ SiOx Ag); (**d**) TiOB^®^ functionalized with ionic zinc (TiOB^®^ Zn).

**Figure 3 materials-12-00866-f003:**
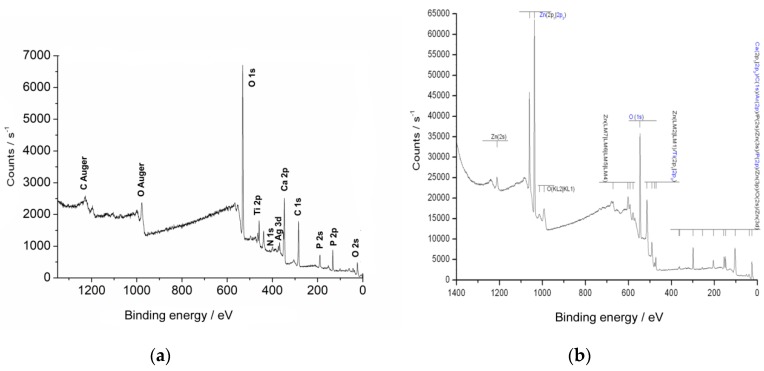
X-ray photoelectron spectroscopic analysis: (**a**) TiOB^®^ SiOx Ag; (**b**) TiOB^®^ Zn.

**Figure 4 materials-12-00866-f004:**
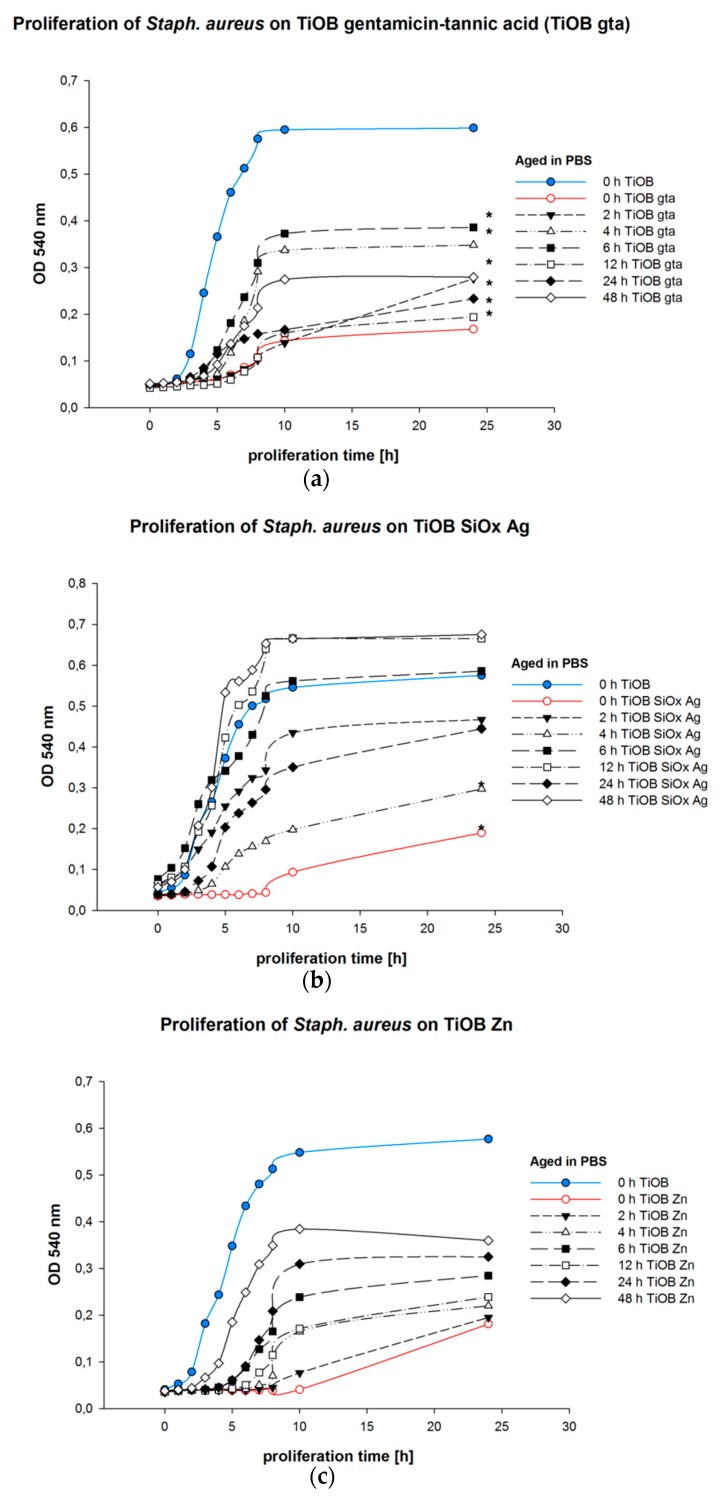
Proliferation of Staphylococcus aureus on functionalized TiOB^®^: (**a**) TiOB^®^ gta; (**b**) TiOB^®^ SiOx Ag; (**c**) TiOB^®^ Zn. Samples were pre-incubated (aged) in PBS for 0, 2, 4, 6, 12, 24, and 48 h. Non-functionalized TiOB^®^ (TiOB^®^ control) is presented as a blue graph. Non-aged but antibacterial functionalized samples are shown in red. Significances to non-functionalized TiOB^®^ are marked by stars (*p* < 0.05).

**Figure 5 materials-12-00866-f005:**
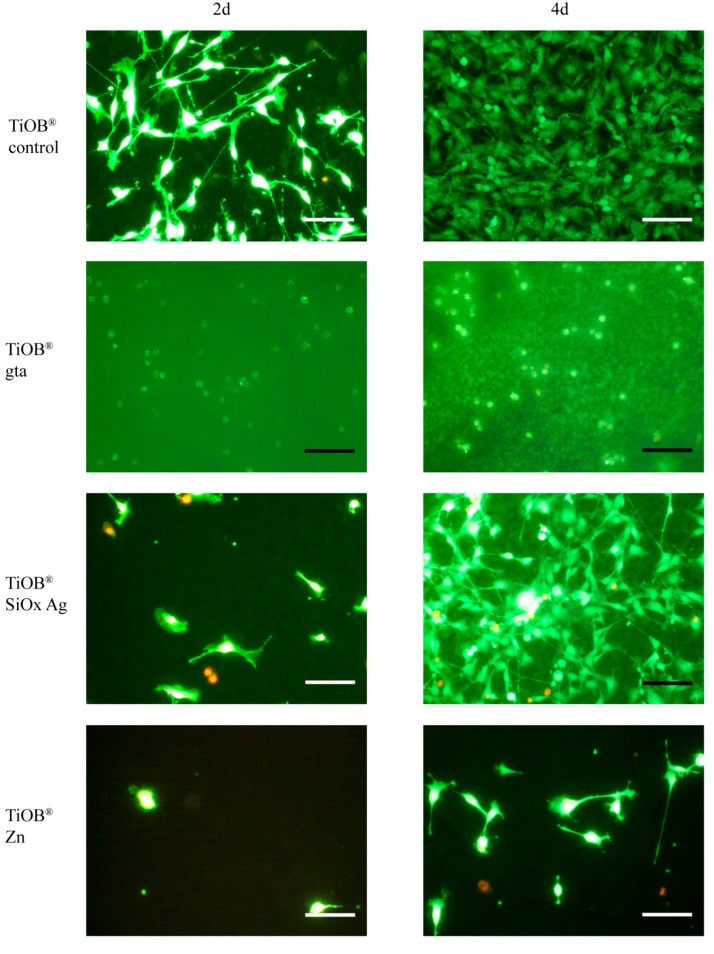
Laser scanning microscopy images of MC3T3-E1 cells cultivated in direct contact with non-functionalized TiOB^®^ (TiOB^®^ control), TiOB^®^ gta, TiOB^®^ SiOx Ag, and TiOB^®^ Zn for 2 and 4 days. Cells were stained by FDA/Ethidium bromide. Viable cells appear in green whereas nuclei of non-viable cells are shown in red.

**Figure 6 materials-12-00866-f006:**
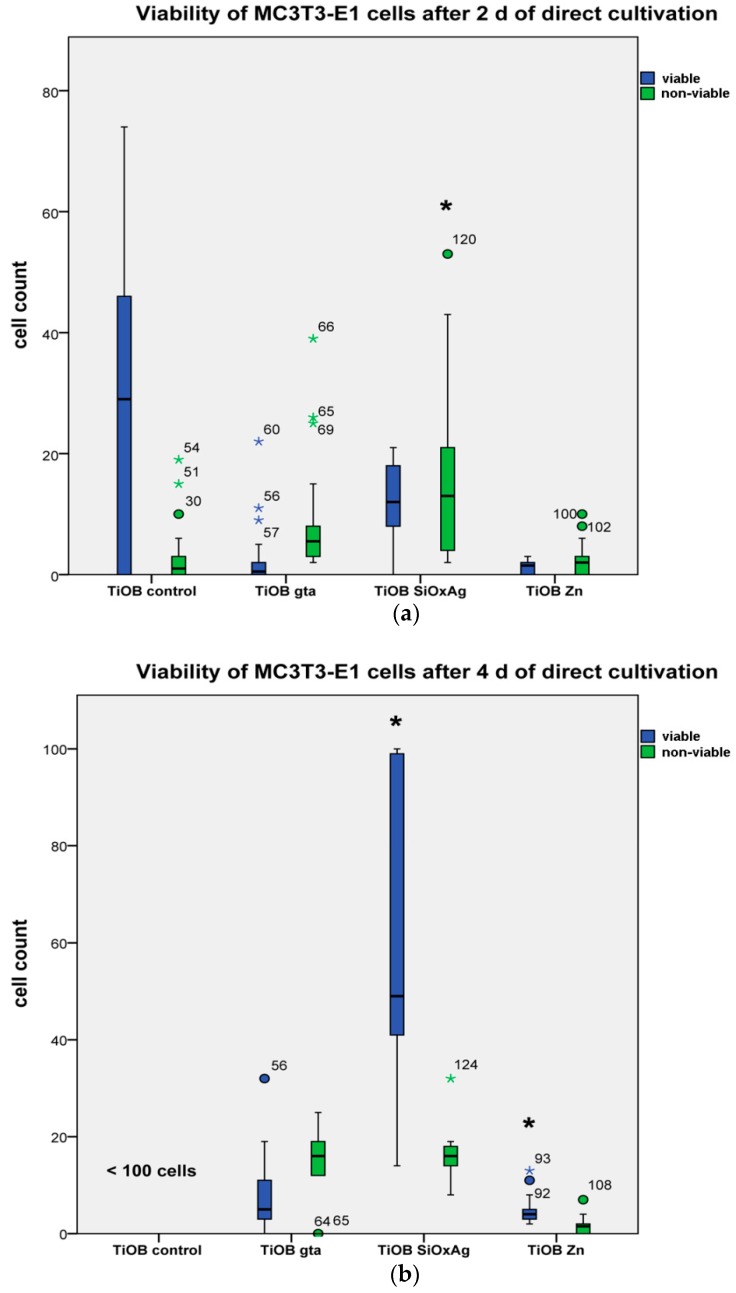
Share of viable and non-viable cells grown in direct contact with non-functionalized TiOB^®^ (TiOB^®^ control), TiOB^®^ gta, TiOB^®^ SiOx Ag, and TiOB^®^ Zn after 2 (**a**) and 4 (**b**) days of cultivation. Significant differences are marked with a star (*p* < 0.05).

**Table 1 materials-12-00866-t001:** Group assignments and technical information.

Group Assignment	Technical Information
TiOB control	Ti6Al4V, PCO (280 V), TiOB surface
TiOB gentamicin-tannic acid (TiOB gta)	Ti6Al4V, PCO (280 V), TiOB surface, dip coating
TiOB SiOx Ag	Ti6Al4V, PCO (280 V), TiOB surface, APCVD
TiOB Zn	Ti6Al4V, 1st PCO (200 V), TiOB surface, 2nd PCO (350 V), Zn electrolyte

**Table 2 materials-12-00866-t002:** Inhibition of *Staphylococcus aureus* in an agar diffusion test by eluates collected from the functionalized surfaces after 0, 2, 4, 6, 12, 24, and 48 h.

	Agar Diffusion Test- -							
	Eluates Collecting Time							
	0 h	2 h	4 h	6 h	12 h	24 h	48 h	
**TiOB^®^ control**	0	0	0	0	0	0	0	**mean inhibition zones [mm]**
**TiOB^®^ gta**	0	1.94	1.88	0.50	0.94	1.50	1.25	
**TiOB8^®^ SiOx Ag**	0	0	0	0	0	0	0	
**TiOB^®^ Zn**	0	0	0	0	0	0	0	

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
