# Peer review of "Bactericidal and Biocompatible Properties of Plasma Chemical Oxidized Titanium (TiOB®) with Antimicrobial Surface Functionalization"

_materials, 2019, doi:10.3390/ma12060866_

Reviewer 1 Report

Although it is a very interesting paper, there are some questions to ask.
1. Is not the material surface observation less likely? Since it is the interface research, evaluation of SPM should also be added. Also, please add narrow scan for XPS.
2. Bacterial experiments are effective, but there are too few bacteria to use. You should add bacteria related to implant implantation.
3. If you observe hard tissue formation around the implant material, you should evaluate ALP and mineralization.
4. Please describe to what extent the effect on the surface is sustained.
5. Will the strength of the material itself be reduced by modifying the material surface?

Author Response

Dear Reviewer,

thank you for your kind comments and suggestions on our manuscript. We tried to answer all of your questions to our best knowledge. All changes among the manuscript are marked in green.

Statement: Is not the material surface observation less likely? Since it is the interface research, evaluation of SPM should also be added. Also, please add narrow scan for XPS.

Answer: Thank you for this remark. The primary focus of this study was to investigate the efficiency of antibacterial coatings on TiOB that are different from the traditional mechanisms of antibiotics. In addition to the antibacterial activity those surfaces should present none or only little cytotoxicity. The reviewer is right telling that the surface topography has a big impact on the biocompatibility and incorporation behavior of implants. But as already mentioned, the main focus of the present study was to evaluate the antibacterial efficiency. Since, incorporation in-vivo as well as osseointegration or bone formation was not observed we will not add any further details upon the surface morphology. As adviced, a narrow XPS scan for TiOB® SiOx Ag was added to the manuscript.  

Statement: Bacterial experiments are effective, but there are too few bacteria to use. You should add bacteria related to implant implantation.

Answer:  Thank you for this comment. Each sample was incubated with 300 µl bacterial solution (OD546 0.5) for one hour at 37 °C under slightly shaking. At an OD546nm of 0.5 a concentration of approximately 106–107 bacterial cells/ml are present in the bacterial solution. For reference please view: Ossmann A, Kranz S, Andre G, Völpel A, Albrecht V, Fahr A, Sigusch BW. Photodynamic killing of Enterococcus faecalis in dentinal tubules using mTHPC incorporated in liposomes and invasomes. Clin Oral Investig. 2015 Mar;19(2):373-84.

We also added this information to the methodical section. Please view page 5, line 181.

As already stated in the text, the proliferation assay was adopted from Bechert et al.. For further information regarding the proliferation assay, please view: Bechert, T.; Steinrucke, P.; Guggenbichler, J.P. A new method for screening anti-infective biomaterials. Nature medicine 2000, 6, 1053-1056.

In cases of implant-associated infections of orthopedic implants, Staphylococcus spp. are considered the predominant bacteria with Staphylococcus aureus often being the main causative agent. Staphylococcus aureus was also applied in the present study and can be found in periimplatitis sites, too. (Lafaurie GI, Sabogal MA, Castillo DM, Rincón MV, Gómez LA, Lesmes YA, Chambrone L. Microbiome and Microbial Biofilm Profiles of Peri-Implantitis: A Systematic Review. J Periodontol. 2017 Oct;88(10):1066-1089).

Statement: If you observe hard tissue formation around the implant material, you should evaluate ALP and mineralization.

Answer: Hard tissue formation was not observed in this study.

Statement: Please describe to what extent the effect on the surface is sustained

Answer: It was shown that TiOB gta is active for up to 24 h (Diefenbeck, M.; Schrader, C.; Gras, F.; Muckley, T.; Schmidt, J.; Zankovych, S.; Bossert, J.; Jandt, K.D.; Volpel, A.; Sigusch, B.W., et al. Gentamicin coating of plasma chemical oxidized titanium alloy prevents implant-related osteomyelitis in rats. Biomaterials 2016, 101, 156-164). All other functionalized surfaces were not yet subjected to any long-term tests. But, we are very thankful for this remark! The long-term effect of TiOB SiOx Ag and TiOB Zn will soon be evaluated in an animal study. In the present study extended lag phases of all non-aged surfaces were found (TiOB® gta for 5 h, TiOB® SiOx Ag for 8 h, TiOB® Zn for 10 h). In addition, TiOB® gta and TiOB® Zn showed a bactericidal effect for 48 h while TiOB® SiOx Ag was active for only 4 h. TiOB alone is very inert and will be degraded in the body only after a very long period of time.

Statement: Will the strength of the material itself be reduced by modifying the material surface?

Answer: The application of a coating will not influence the strength of the material itself. During the application process temperatures of about 50°C were reached which does not influence the chemical and physical properties of the used TiAI6V4 alloy. On page 3, line 121 there is a mistake. All coatings were applied at room temperature and not as stated at temperatures of 500-700°C. We corrected this mistake! Thank you for this comment!

Thank you for reviewing our manuscript and your comments which helped to improve the scientific appearance of the paper!

Reviewer 2 Report

The paper is very well thought out and can be considered an interdisciplinary one. In my opinion, it is ready for printing.

Author Response

Dear reviewer,

thank you for reviewing our manuscript and your kind comments!

Reviewer 3 Report

This is an interesting study focused on analyze biologica properties of bioactive TIOB functionalized  with different additives.

Some criticism are present :

-Can authors indicate in FIgure1 sections a and b?

-Can authors indicate why sample dimensions were chosen?

-The authors should add in introduction or discussion section some considerations about citotoxicyty of different dental materials to oral cells to accent paper novelty.

IN this context i can add some references about this problem:

-Marinucci L, Balloni S, Bodo M, Carinci F, Pezzetti F, Stabellini G, Conte C Lumare E "Patterns of some extracellular matrix gene expression are similar in cells from cleft lip-palate patients and in human palatal fibroblasts exposed to diazepam in culture Toxicology, Vol. 257 (1-2), pag 10-16, 2009.

-Pagano, S, Chieruzzi M, Balloni S, Lombardo G, Torre L, Bodo M, Cianetti S, Marinucci L "Biological, thermal, mechanical characterization of modified glass ionomer cements: The role of nanohydroxyapatite, ciprofloxacin and Zin L-carnosine "Mater SCI eng C mater biol Appl. 2019 Jan 1;94:76-85. 

Author Response

Dear Reviewer,

thank you for your comments and suggestions. We tried to answer your questions to our best knowledge. All changes among the manuscript are marked in red.

Statement: Can authors indicate in FIgure1 sections a and b?

Answer: We indicated in Figure 1 section a and b. Also, we introduced a new caption for the figure. Thank you for the advice!

Statement: Can authors indicate why sample dimensions were chosen?

Answer: The proliferation test and sample design was adopted from Bechert et al. For more detailed information please view: Bechert, T.; Steinrucke, P.; Guggenbichler, J.P. A new method for screening anti-infective biomaterials. Nature medicine 2000, 6, 1053-1056. The disc shape samples used in the cytotoxicity tests were manufactured by Königseeimplantate GmbH. These samples were of round shape with 15 mm in diameter and 2 mm in thickness. The design was chosen in order to ensure sufficient handling in the cell culture and microscopy procedures.    

 Statement: The authors should add in introduction or discussion section some considerations about citotoxicyty of different dental materials to oral cells to accent paper novelty.

 IN this context i can add some references about this problem:

 -Marinucci L, Balloni S, Bodo M, Carinci F, Pezzetti F, Stabellini G, Conte C Lumare E "Patterns of some extracellular matrix gene expression are similar in cells from cleft lip-palate patients and in human palatal fibroblasts exposed to diazepam in culture Toxicology, Vol. 257 (1-2), pag 10-16, 2009.

 -Pagano, S, Chieruzzi M, Balloni S, Lombardo G, Torre L, Bodo M, Cianetti S, Marinucci L "Biological, thermal, mechanical characterization of modified glass ionomer cements: The role of nanohydroxyapatite, ciprofloxacin and Zin L-carnosine "Mater SCI eng C mater biol Appl. 2019 Jan 1;94:76-85.

 Answer: Thank you for this comment. We added additional information to the manuscript. Please view page 13, line 341 and page 14, line 392. Additional references were introduced which caused some changes in the reference section, too.

 Thank you for reviewing our manuscript and for the kind comments which helped to improve the scientific quality of the paper! 

 Round  2

Reviewer 1 Report

I think this apply is very good.